# Transboundary Air Pollution Transport of PM$_{10}$ and Benzo[*a*]pyrene in the Czech–Polish Border Region

Vladimíra Volná *, Daniel Hladký, Radim Seibert and Blanka Krejčí

Ambient Air Quality Department, Czech Hydrometeorological Institute, 708 00 Ostrava, Czech Republic; daniel.hladky@chmi.cz (D.H.); radim.seibert@chmi.cz (R.S.); blanka.krejci@chmi.cz (B.K.)
* Correspondence: vladimira.volna@chmi.cz

**Abstract:** The article is occupied with the evaluation of the transboundary transport of pollutants in the Czech–Polish border region (between the Moravian-Silesian region and the Silesian Voivodeship) in Central Europe. It focuses on the evaluation of concentrations of benzo[*a*]pyrene (BaP) and suspended particles of PM$_{10}$ depending on meteorological conditions, especially wind direction. The whole area of interest is heavily affected with air pollution of BaP and PM$_{10}$. Limits of BaP and PM$_{10}$ are still exceeded. Annual concentrations of benzo[*a*]pyrene are even several times higher than the value of its annual limit. The elaboration follows the results of the Czech–Polish project "Air Silesia", which dealt with air pollution and the transboundary transport of pollutants in this area and took place in 2010 to 2013. Within this project, a higher transport of pollutants from Poland to the Czech Republic was established. The evaluation of the dependences of PM$_{10}$ concentrations is based on hourly and daily data of PM$_{10}$ and hourly data of meteorological quantities. To assess the dependences of daily BaP and PM$_{10}$ concentrations, a methodology for evaluating daily types of wind direction was implemented into the processing. The results confirm that the problem of above-limit concentrations of BaP and PM$_{10}$ in the Moravian-Silesian Region in the Czech Republic and the Silesian Voivodeship in Poland remains. The article confirms there is a higher transport of PM$_{10}$ concentrations from Poland to the Czech Republic in the area of interest. Higher transport in the same direction is also predicted for daily concentrations of benzo[*a*]pyrene, although this cannot be clearly confirmed due to the lack of more detailed and identifiable data.

**Keywords:** Czech–Polish border; transboundary air pollution transport; PM$_{10}$; benzo[*a*]pyrene; daily type of wind

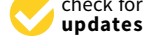



## 1. Introduction

Suspended particles PM$_{10}$ and benzo[*a*]pyrene are problematic pollutants and have clinically proven negative effects on the human body, especially in the form of carcinogenicity [1–5]. The area of interest is located in the region where the limits for both pollutants are being exceeded in the largest area in the whole of Europe [6]. The entire Czech–Polish region is part of the Upper Silesian Basin, where coal mining has taken place since the 17th century. The area of interest is characterised by a high density of industrial resources, transport infrastructure and a high density of settlements with individual heating with solid fuels. The negative impact of the above-mentioned components on the environment and air quality has been recorded since the 1970s. Improvements in air quality occurred in the 1990s with the advent of economic restructuring in both post-communist countries [7,8]. The accession of both countries to the European Union and the need to adapt legislation to EU requirements has also contributed significantly to the improvement [9,10]. Thanks to these measures, air pollution levels in the Czech–Polish region have been reduced but problems persist, and there are still exceedances of the limits for PM$_{10}$ and benzo[*a*]pyrene.

In addition to the parameters of pollution sources and the amount of pollutants emitted, meteorological conditions influence the level of air quality. Meteorological conditions

determine the intensity and dispersion of pollutants in the air [11,12]. The most important and decisive dispersion conditions are considered to be the wind direction and speed and the vertical stability of the atmosphere, conditioned by the air temperature. Wind characteristics influence the transport of pollutants in the horizontal direction, while atmospheric stability and air temperature influence the dispersion of pollutants in the vertical direction. In the most stable situations, air temperature increases with height and conditions for vertical mixing are least favourable. Conversely, under unstable stratification, temperature decreases with height faster than would be consistent with normal atmospheric conditions and conditions for pollutant dispersion are favourable [13–22].

The transboundary transport of pollutants across state borders is a very complex and sensitive issue. This is due, among other things, to the often complex course of the national border, the temporal variability of the frequency of wind across the border (both in terms of direction and wind speed), the temporal variability of the vertical stability of the near-surface layers of the atmosphere, the different distribution and nature of emission sources on both sides of the border (low/high), and the long-range transport of pollutants in the atmosphere. As a consequence, the transboundary transport of pollutants across a certain section of the national border can only be estimated. Either mathematical modelling methods of the atmospheric dispersion of pollutants or direct measurements of the pollutants and necessary additional meteorological measurements along the border line can be used for estimation and the dependence of measured concentrations on the direction of wind can be studied. In this way, however, it is only possible to assess transboundary transport at the level of measurements at the surface, not the transport of pollutants in the upper atmosphere [23–25]. For these purposes, distance or flight measurements are required. The content of this paper is an attempt to estimate the quantification of the transboundary air pollution transport between the Czech Republic and Poland based on the evaluation of $PM_{10}$ and benzo[*a*]pyrene (BaP) concentrations as a function of wind direction and wind speed.

The evaluation of meteorological–emission relations and the transboundary transport of pollutants in the Czech–Polish border area of interest was previously addressed by the authors within the project "Air Quality Information System in the Polish-Czech border in the Silesian and Moravian-Silesian Region" (Air Silesia), financed by the Operational Programme for Czech–Polish Cross-border Cooperation 2007–2013 (registration number of the project: CZ.3.22/1.2.00/09.01610; the sustainability period of the Air Silesia project has ended; no website already available). On the basis of previous work, it was found that the influence of wind speed and vertical temperature stratification on air pollutant concentrations is dominant, the influence of wind direction in the area is complementary. The exceptionally high concentrations are due to prolonged inversion situations throughout the region during the winter period, i.e., from December to February. The inter-annual variability in pollutant concentrations is high and depends on meteorological conditions, especially in the cold half of the year. The wind direction is predominantly from the southwest and is typical for northeastern Moravia and is related to the orographic influence of the Moravian Gate. This influence is also evident in the adjacent parts of the Silesian Voivodeship. Under predominantly good dispersion conditions, pollutants are mostly transported from the Moravian-Silesian region to the Silesian Voivodeship region, whereas the opposite is true under predominantly poor dispersion conditions. Sites located in the central part of the cross-border area are polluted in both wind directions by emissions from sources located in the downwind part of the area [23–25].

In the present paper, the authors attempt to build on the previous treatment [23–25]. The assessment will be based on the 10-year treatment period 2011–2020 for suspended $PM_{10}$ as a function of wind direction and wind speed. For the assessment of daily pollutant concentrations depending on wind direction, daily flow types were calculated according to the new methodology used by the Czech Hydrometeorological Institute (CHMI, Czech Republic). Available daily BaP concentrations will also be included in the processing.

The results of the processing are a valuable basis for bilateral negotiations between the state administration authorities of both the Czech Republic and Poland. The assessment can help to take better measures and agreements between the two countries and, thus, contribute to improving the level of air pollution in the border area. There are few works dealing with transboundary air pollution transport in the area of interest. In particular, these are works linked to international projects in the area. However, the sub-projects tend to deal with the measurement and assessment of air pollution at the project sites and do not address the transport of pollutants in the Czech–Polish region [26,27].

## 2. Experiments

Two stations located close to the Czech–Polish border were selected for evaluation (Figures 1 and 2). The Czech station Věřňovice [28] is a station of the Czech Hydrometeorological Institute (CHMI) and is located at an altitude of 203 m less than 800 m from the state border with Poland. The Polish station Godów [29] is a station of the Chief Inspectorate of Environmental Protection (GIOŚ, Poland) and is located at 204 m less than 500 m from the state border with the Czech Republic. The monitored sites are about 3.5 km apart in a straight line, in the west–east axis. Both sites belong to the National Air Quality Monitoring Network of Czech republic [30] and Poland. At the Věřňovice station, pollutants and meteorological variables are measured in an automatic programme (1-h data). Godów is a manual station (24-h data). The data from the Věřňovice station were supplied by the Czech Hydrometeorological Institute for this processing. Data from the Godów station are managed by the Chief Inspectorate of Environmental Protection (GIOŚ) and are available on their website (https://powietrze.gios.gov.pl/pjp/archives, accessed on 10 January 2022). The evaluation includes 1-h concentrations of suspended $PM_{10}$, temperature and wind direction and speed from the Věřňovice station, and 24-h concentrations of $PM_{10}$ from the Godów station. Benzo[*a*]pyrene concentrations could only be included in the assessment from the Věřňovice station and the year 2020, when measurements were taken every 6th day. In other years, no BaP measurements were carried out in Věřňovice. Although BaP was measured at Godów for the entire assessment period 2011–2020, it could not be fully included in the treatment. Daily BaP concentrations are not related to a specific measurement day but are budgeted for each day of the week (Monday to Sunday). One concentration is therefore available for the whole week (Monday to Sunday), but it does not have an identifier for which day it was measured (https://powietrze.gios.gov.pl/pjp/archives, accessed on 10 January 2022). Annual BaP concentrations can be calculated, but daily concentrations cannot be evaluated in relation to other variables, e.g., wind direction and wind speed. The measurements of meteorological variables at the Věřňovice station are representative of the wider area [28], and, therefore, the wind direction and wind speed (or temperature) data can be used for the Godów station. The processing period is 2011 to 2020 with the limitations described above.

At the Věřňovice site, $PM_{10}$ concentrations are measured by means of a radiometric method based on the absorption of beta radiation in a sample collected on filter material by means of an MP101M machine (Environnement S.A, Poissy, France) [31]. The wind direction and speed are measured at a standard height of 10 m by means of a WindSonic machine (Gill Instruments Limited, Lymington, United Kingdom), based on ultrasound technology. The air temperature is scanned at a height of 2 m above the ground, by means of a Cormet company machine. In Godów, $PM_{10}$ is measured by gravimetry. Concentrations of benzo[*a*]pyrene are measured as $PM_{10}$ content using a SEQ 47/50 sequential cooled sampler (Sven Leckel, Berlin, Germany). Furthermore, the samples are analysed in the laboratories using gas chromatography with mass selective detection [31].

In the analysis, concentration roses were also used for illustrative purposes, which show average $PM_{10}$ concentrations for the given wind direction and speed, as well as weighted concentration roses [32]. The difference between a concentration rose and a weighted concentration rose is that the latter provides information about how often a given wind direction and speed combination occurs and states to what extent the concentrations

detected for a given wind speed and direction affect the overall average concentration for a given period. The comparison of the two roses may show a significant pollution source located, however, in a sector from which the wind only rarely blows and thus does not contribute significantly to the overall average concentration. The concentration rose reveals what the pollution situation was at maximum concentrations of a given contaminant at a particular site; the weighted concentration rose shows from what wind direction and at what speed the pollution came to the largest extent for the whole period. For the above-described reasons, both rose types for the same site and period may vary significantly [33]. In the presented roses, there is information about calm air, which is a situation with a wind speed of 0–0.2 m·s$^{-1}$.

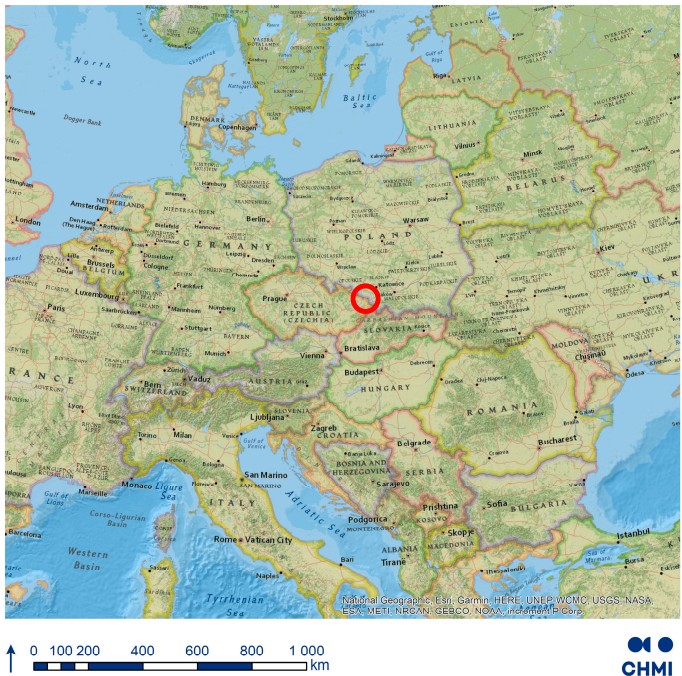

**Figure 1.** The assessed site location within Central Europe.

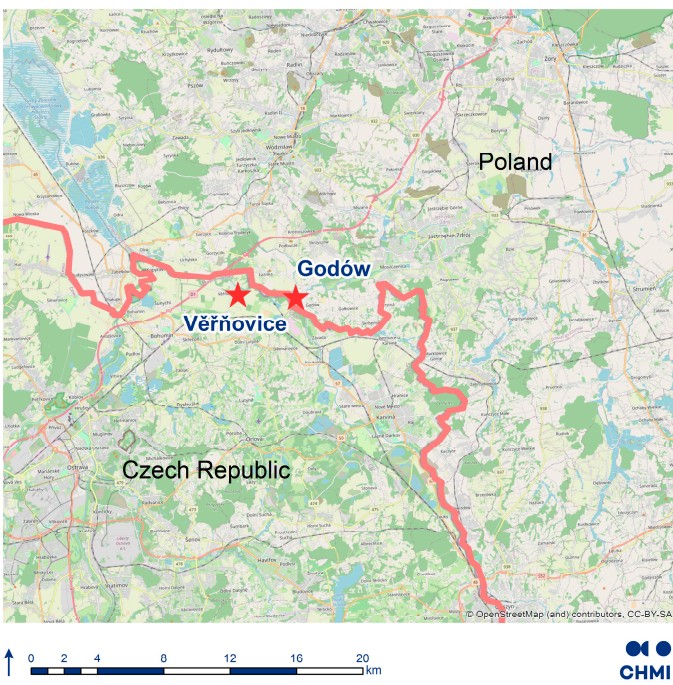

**Figure 2.** The assessed site location within the Czech–Polish border region.

For the assessment of 24-h pollutant concentrations depending on wind direction, the internal methodology of the CHMI for determining the daily type of wind direction was used. The methodology determines the daily prevailing wind direction, which is calculated from hourly wind direction and wind speed data, and at least 18 values must be available on a given day. Based on the methodology, it is possible to identify 8 basic directions N, NE, NW, W, S, SW, E, SE (North, North-East, North-West, West, South, South-West, East, South-East), or sectors for two directions located next to each other (N–SW, N–SW, N–SW, N–SW, N–SW, N–SW, NW–W), as well as for no wind ($\leq$0.4 m$\cdot$s$^{-1}$), variable wind and daily type of wind, which could not be specified (variable). The assessment of pollutant concentrations as a function of daily type of wind may not be objective. For example, for BaP at the Věřňovice site, measurements were not made every day. Measurements every 6th day do not objectively include situations with different occurrences of daily benzo[a]pyrene concentrations during the year (either with high or low concentrations), nor is it possible to include the occurrence of all types of meteorological conditions. This is an indicative characteristic with limited predictive value.

To assess the transboundary transport of pollutants in the area of interest between the Czech Republic and Poland, the wind directions (daily types of wind) were divided into two parts according to the axis of the national border. The transport from the Czech Republic to Poland includes the following daily types: W, SW, S, SW–W, S–SW. The transmission from Poland to the Czech Republic in this area includes: NW, N, NE, E, N–NE, NE–E, NW–N. Other wind directions that cannot be clearly classified as transmission directions from the Czech Republic to Poland (and vice versa) are marked as unclassifiable. This classification for the assessment of transboundary transport has been previously used and published [23–25].

Věřňovice is situated in the area of the Ostrava Basin where the "Moravian Gate" ends; this divides the Podbeskydská hilly area and Nízký Jeseník [34,35]. The shape of the gate along the southwest/northeast axis significantly determines the prevailing wind direction of the region where Věřňovice is situated (Figure 3). It is apparent from the existing analyses that local fireplaces have the greatest contribution to the exceedance of PM$_{10}$ pollution limits here among the primary sources, mostly those used on the territory of Poland [36,37].

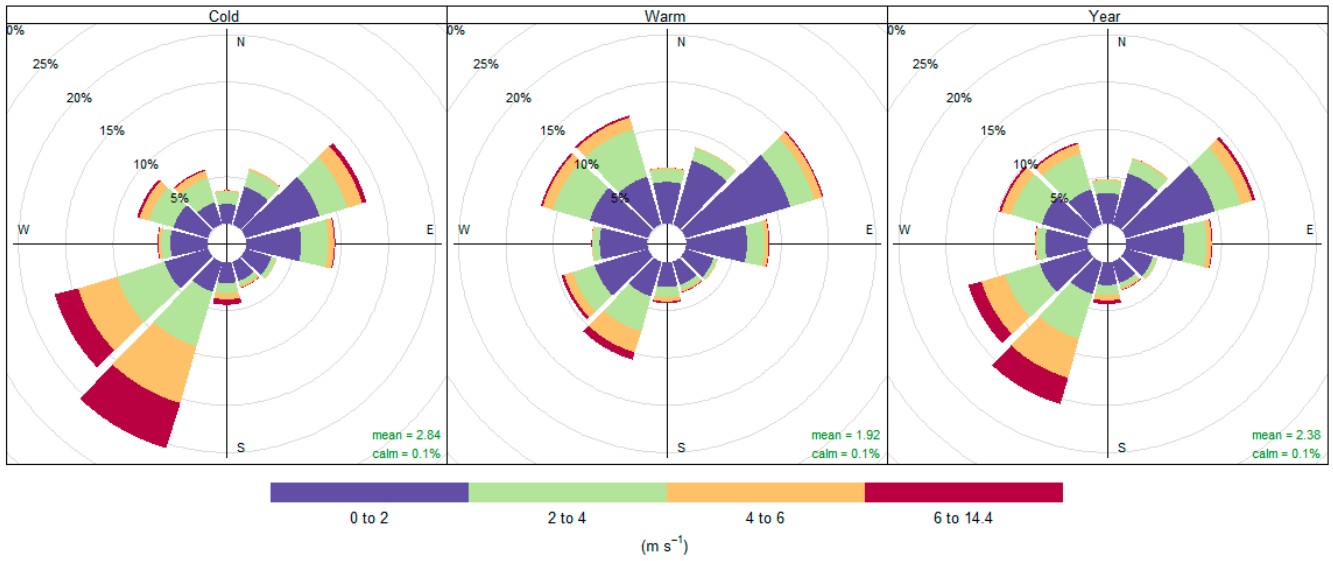

**Figure 3.** Average frequency of counts by wind direction, 2011–2020, Věřňovice.

## 3. Results

In this section the results of PM$_{10}$ and BaP treatment for Godów and Věřňovice stations are presented. The assessment of PM$_{10}$ concentrations for both stations is more detailed in view of the facts described in Section 2 (Experiments).

The quantification of wind direction for the period 2011–2020 representing the direction from the Czech Republic to Poland and from Poland to the Czech Republic is shown in Table 1. In the cold part of the year (months October to March 2011 to 2020), the directions from the Czech Republic to Poland are significantly more prevalent, 46% to 30%. In the warm part of the period 2011–2020, the predominant direction is the opposite, from Poland to the Czech Republic, 41% to 22%. On average over the whole period, the ratio is balanced. The share of variable winds is significant, about 30%. The share of unaligned wind directions for the assessment of pollutant transport between the Czech Republic and Poland is negligible.

**Table 1.** Quantification of wind directions between the Czech Republic and Poland, 2011–2020.

|  | Cold Season 2011–2020 (%) | Warm Season 2011–2020 (%) | 2011–2020 (%) |
|---|---|---|---|
| Czechia → Poland | 46 | 22 | 34 |
| Poland → Czechia | 30 | 41 | 36 |
| Variable | 22 | 35 | 29 |
| Unclassifiable | 2 | 1 | 2 |

### 3.1. Evaluation of PM$_{10}$ Concentrations

The development of average PM$_{10}$ concentrations at both stations is very similar (Figure 4). The lowest average daily and annual concentrations were found in 2020 and the lowest annual mean and median were at the Godów station. The highest annual mean PM$_{10}$ concentration was achieved in 2012 at the Věřňovice station (56 µg·m$^{-3}$) and at the Godów station in 2014 (50 µg·m$^{-3}$). Maximum daily concentrations were detected at Věřňovice in 2012 (548 µg·m$^{-3}$) and Godów in 2017 (495 µg·m$^{-3}$). Between 2011 and 2020, the annual average PM$_{10}$ concentrations were above or just above the annual limit of 40 µg·m$^{-3}$. The exception is 2020, when the average annual concentration in Věřňovice was 28 µg·m$^{-3}$ and in Godów 31 µg·m$^{-3}$.

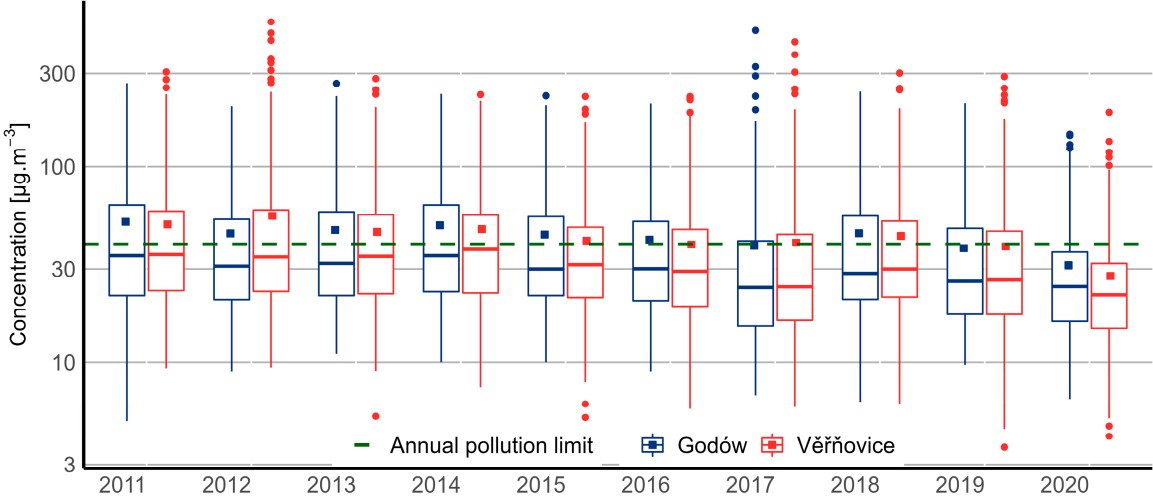

**Figure 4.** Boxplot of daily PM$_{10}$ concentrations, Godów and Věřňovice stations, 2011–2020 (blue and red squares indicate average concentrations).

To assess the dependences of the behaviour of concentrations at the stations Věřňovice and Godów, it was possible to use (see the description in Section 2) only daily concentrations of PM$_{10}$ for the period 2011 to 2020. The results in Figure 5 show a significant correlation dependence between the compared stations. The summer season (R$^2$ = 0.62) shows the least favourable results, the dependence for cold seasons (R$^2$ = 0.88) is significantly better.

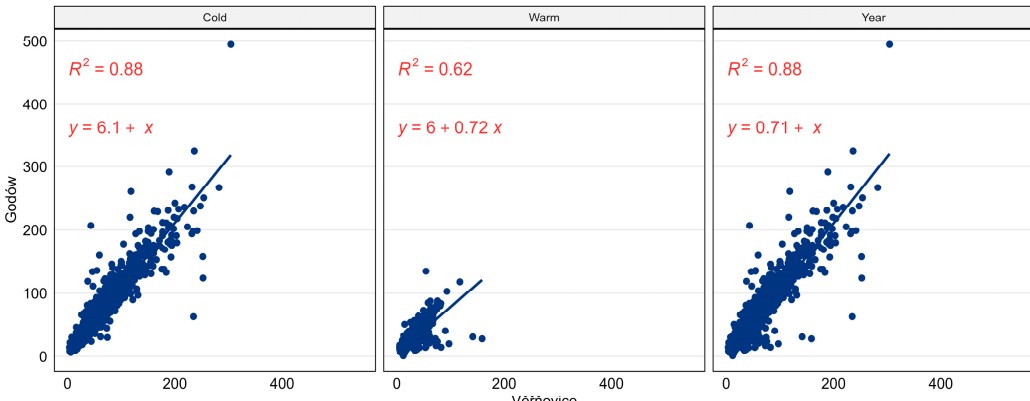

**Figure 5.** Regression dependences between average daily $PM_{10}$ concentrations, Godów and Věřňovice stations, 2011–2020 [μg·m$^{-3}$].

The weighted concentration roses for $PM_{10}$ at the Věřňovice site (Figures 6 and 7) show that the most frequent average contributions of $PM_{10}$ concentrations come at the station from the NE to E direction at wind speeds up to 2 m·s$^{-1}$. The highest $PM_{10}$ concentrations are obtained in the cold months of the year. More illustrative is Figure 7, which does not include wind speed information. The maximum 1-h contributions at the Věřňovice station (Figure 8) come from the NE and up to E directions at low wind speeds around 1 m·s$^{-1}$. The SW direction is the predominant flow direction at the station, but lower average $PM_{10}$ concentrations come from this direction. High contributions of maximum short-term $PM_{10}$ concentrations are also not recorded from the SW direction.

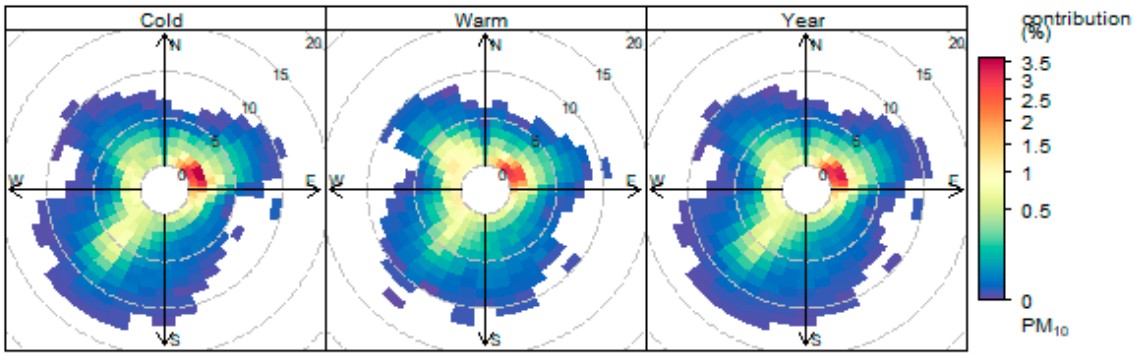

**Figure 6.** $PM_{10}$ weighted concentration rose, the Věřňovice site, 2011–2020.

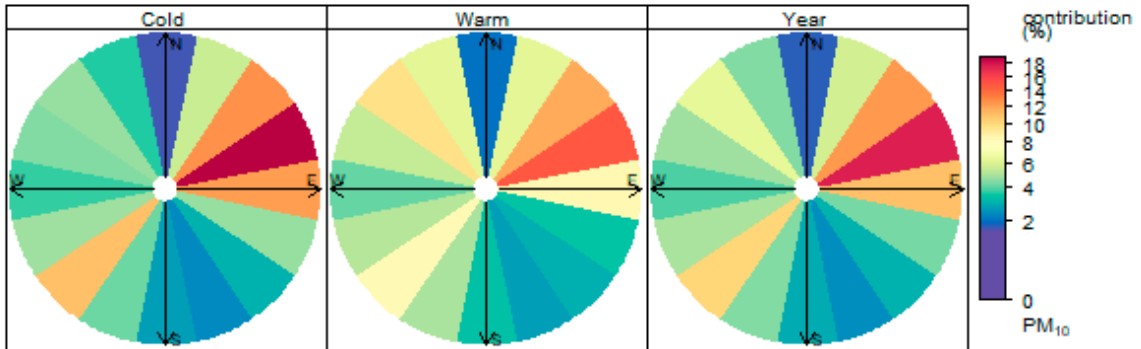

**Figure 7.** $PM_{10}$ weighted concentration rose (except for the wind speed parameter), the Věřňovice site, 2011–2020.

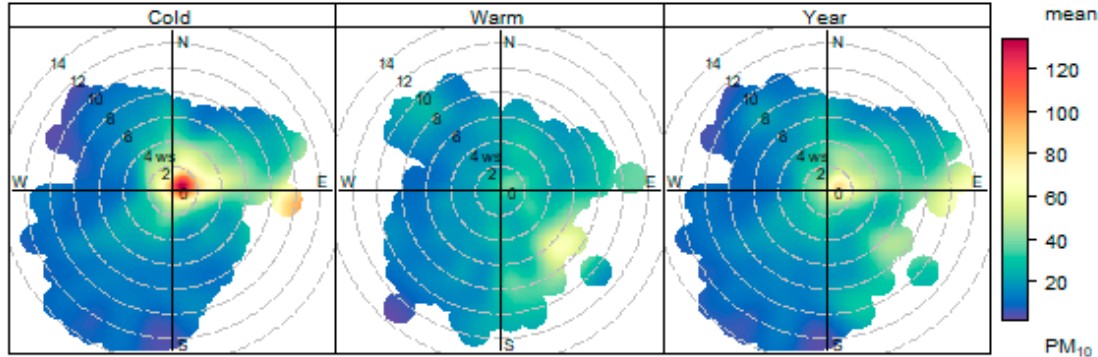

**Figure 8.** $PM_{10}$ concentration roses, Věřňovice site, 2011–2020.

The average $PM_{10}$ concentrations for each daily type of wind in the cold seasons 2011–2020 (Figure 9) show that the effect of wind direction is similar at both stations. The highest average $PM_{10}$ concentrations in the cold seasons of 2011–2020 are clearly obtained for the NE–E daily type of wind, followed by the E and variable wind direction. The average $PM_{10}$ concentrations are also highest in the cold part of the year. The average $PM_{10}$ concentrations in the cold months are 2–4 times higher than the average concentrations in the warm months of 2011–2020 (Figure 10). The average $PM_{10}$ concentrations for the entire 2011–2020 period (Figure 11) show that the highest $PM_{10}$ concentrations are obtained when the daily wind type is E, NE–E or N–NE.

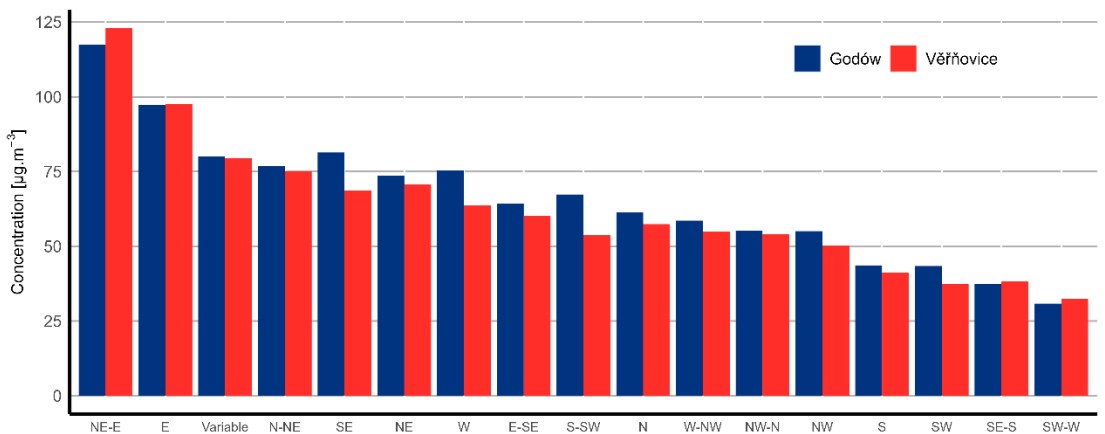

**Figure 9.** Average $PM_{10}$ concentrations for individual daily wind types, Věřňovice and Godów site, cold seasons (X–III) 2011–2020.

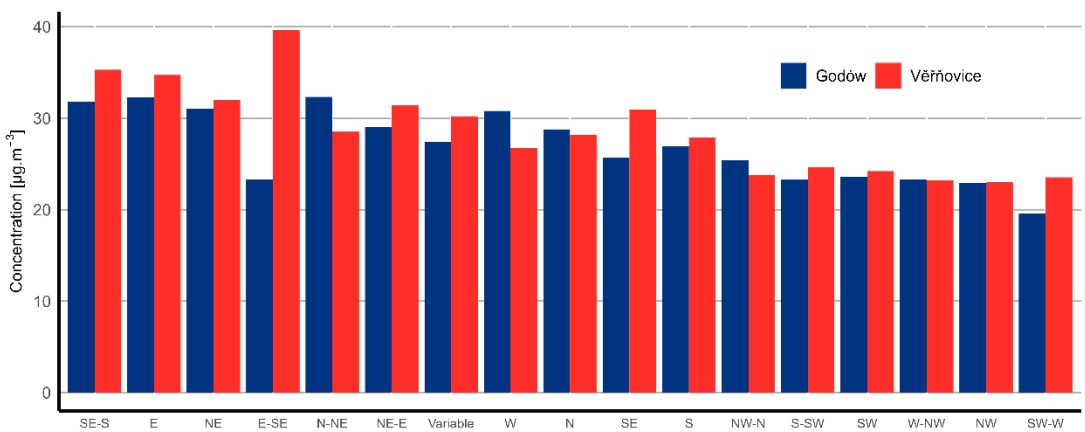

**Figure 10.** Average $PM_{10}$ concentrations for individual daily wind types, Věřňovice and Godów site, warm seasons (I–IX) 2011–2020.

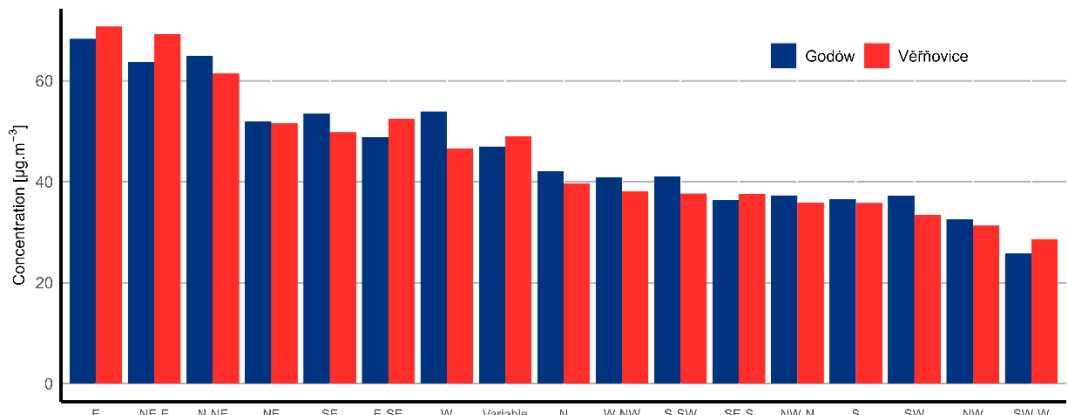

**Figure 11.** Average $PM_{10}$ concentrations for individual daily wind types, Věřňovice and Godów site, 2011–2020.

The dependence of $PM_{10}$ concentrations on temperature is significant [38,39]. Figure 12 shows the $PM_{10}$ concentrations as a function of temperature and divided into individual daily types of wind. At low temperatures below 1 °C, the highest $PM_{10}$ concentrations (averaging more than 150 μg·m$^{-3}$) are obtained in the NE–E wind direction. At temperatures of 1–5 °C the influence of the E, SE and W directions is evident. At temperatures above 10 °C, the contribution of wind directions to $PM_{10}$ concentrations is more balanced, with the lowest concentrations at SW–W.

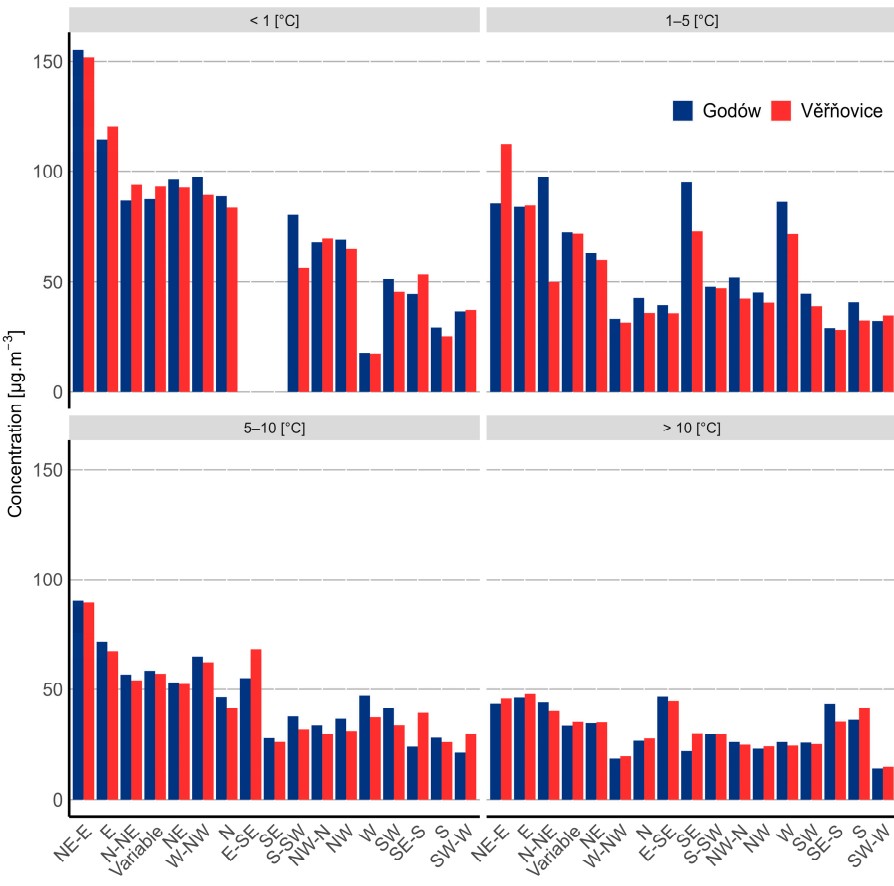

**Figure 12.** Average $PM_{10}$ concentrations by individual daily types of wind and temperature, Věřňovice and Godów site, 2011–2020.

The dependence of $PM_{10}$ concentrations on wind speed is significant [38,39]. Figure 13 shows the $PM_{10}$ concentrations as a function of wind speed and divided into individual daily types of wind. At low wind speeds of 0.5–1 m·s$^{-1}$, the highest average $PM_{10}$ concentrations (around 100 μg·m$^{-3}$) are achieved in the E–SE and NE–E daily types of wind. At wind speeds of 1–2 m·s$^{-1}$, the highest average $PM_{10}$ concentrations are found at N–NE, E and W, and at wind speeds above 2 m·s$^{-1}$ at the SE wind direction.

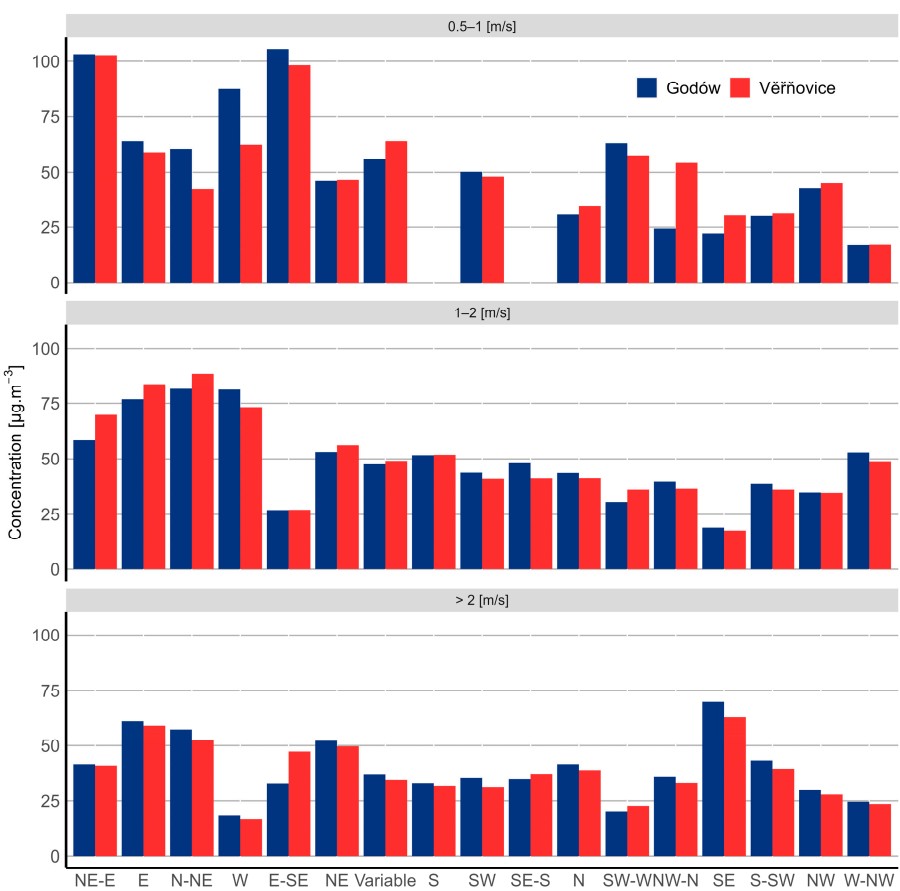

**Figure 13.** Average $PM_{10}$ concentrations by individual daily wind types and wind speed, Věřňovice and Godów site, 2011–2020.

The quantification of the transboundary transport of average $PM_{10}$ concentrations is shown in Table 2. In the warm season, the transport of $PM_{10}$ is almost balanced in both directions, slightly exceeding from Poland to the Czech Republic. In the cold season, $PM_{10}$ transmission from Poland to the Czech Republic is predominant. In the opposite direction from the Czech Republic to Poland, it is about 15% less, with variable wind contributing about 30% to the resulting concentrations. The average $PM_{10}$ concentration (78 μg·m$^{-3}$ at Godów, 77 μg·m$^{-3}$ at Věřňovice) of the cold seasons of the whole period 2011 to 2020 from Poland to the Czech Republic is higher than the value of the daily $PM_{10}$ limit of 50 μg·m$^{-3}$. On average over the whole period, the transfer from Poland to the Czech Republic is higher by 6 to 8%, with average concentrations at both stations in the direction from Poland to the Czech Republic of 48 μg·m$^{-3}$. If we compare transport from the Czech Republic to Poland, from Poland to the Czech Republic, variable wind and unclassified directions, the transport from Poland to the Czech Republic together with variable wind contributes most to the resulting average $PM_{10}$ concentration, followed by unclassified wind directions and, finally, the transport from the Czech Republic to Poland. The largest differences between contributions to the average concentration at the sites are in the cold part of the year when the highest values of average concentrations are achieved.

**Table 2.** Quantification of transboundary PM$_{10}$ transmission between the Czech Republic and Poland (average PM$_{10}$ concentrations and %), 2011–2020.

| | Cold Season 2011–2020 | | | | Warm Season 2011–2020 | | | | 2011–2020 | | | |
|---|---|---|---|---|---|---|---|---|---|---|---|---|
| | Godów (μg·m$^{-3}$) | Godów (%) | Věřňov. (μg·m$^{-3}$) | Věřňov. (%) | Godów (μg·m$^{-3}$) | Godów (%) | Věřňov. (μg·m$^{-3}$) | Věřňov. (%) | Godów (μg·m$^{-3}$) | Godów (%) | Věřňov. (μg·m$^{-3}$) | Věřňov. (%) |
| Czechia → Poland | 44 | 17 | 38 | 16 | 24 | 23 | 25 | 23 | 37 | 21 | 34 | 20 |
| Poland → Czechia | 78 | 30 | 77 | 31 | 27 | 26 | 27 | 25 | 48 | 27 | 48 | 28 |
| Variable | 80 | 31 | 79 | 32 | 27 | 27 | 30 | 28 | 47 | 27 | 49 | 28 |
| Unclassifiable | 56 | 22 | 53 | 21 | 24 | 24 | 27 | 25 | 43 | 24 | 41 | 24 |

The assessment of the transboundary transport of PM$_{10}$ concentrations in terms of the number of daily average concentrations above 50 and 100 μg·m$^{-3}$ is shown in Tables 3 and 4. The highest numbers of exceedances of the values 50 and 100 μg·m$^{-3}$ are at the wind direction from Poland to the Czech Republic and, furthermore, at the variable wind. The differences are more noticeable just in the exceedances of high daily concentrations of 100 μg·m$^{-3}$ in the cold seasons and in the average of the whole period 2011–2020. The number of daily PM$_{10}$ concentrations exceeding the value of 100 μg·m$^{-3}$ is more than 20% higher for the transmission from Poland to the Czech Republic and for variable winds than for the wind direction from the Czech Republic to Poland.

**Table 3.** Quantification of transboundary PM$_{10}$ transmission between the Czech Republic and Poland (number of exceedances of the value of 50 μg·m$^{-3}$ for average daily PM$_{10}$ concentrations), 2011–2020.

| | Cold Season 2011–2020 | | | | Warm Season 2011–2020 | | | | 2011–2020 | | | |
|---|---|---|---|---|---|---|---|---|---|---|---|---|
| | Godów | Godów (%) | Věřňov. | Věřňov. (%) | Godów | Godów (%) | Věřňov. | Věřňov. (%) | Godów | Godów (%) | Věřňov. | Věřňov. (%) |
| Czechia → Poland | 245 | 30 | 198 | 25 | 12 | 14 | 19 | 16 | 257 | 28 | 217 | 24 |
| Poland → Czechia | 334 | 41 | 330 | 42 | 40 | 45 | 43 | 36 | 374 | 41 | 373 | 41 |
| Variable | 231 | 28 | 251 | 32 | 36 | 41 | 56 | 47 | 267 | 29 | 307 | 34 |
| Unclassifiable | 12 | 1 | 10 | 1 | 0 | 0 | 2 | 2 | 12 | 1 | 12 | 1 |

**Table 4.** Quantification of transboundary PM$_{10}$ transmission between the Czech Republic and Poland (number of exceedances of the value of 100 μg·m$^{-3}$ for average daily PM$_{10}$ concentrations), 2011–2020.

| | Cold Season 2011–2020 | | | | Warm Season 2011–2020 | | | | 2011–2020 | | | |
|---|---|---|---|---|---|---|---|---|---|---|---|---|
| | Godów | Godów (%) | Věřňov. | Věřňov. (%) | Godów | Godów (%) | Věřňov. | Věřňov. (%) | Godów | Godów (%) | Věřňov. | Věřňov. (%) |
| Czechia → Poland | 58 | 20 | 42 | 16 | 0 | 0 | 0 | 0 | 58 | 19 | 42 | 15 |
| Poland → Czechia | 124 | 42 | 128 | 47 | 1 | 33 | 0 | 0 | 125 | 42 | 128 | 47 |
| Variable | 109 | 37 | 95 | 35 | 2 | 67 | 3 | 100 | 111 | 37 | 98 | 36 |
| Unclassifiable | 6 | 2 | 6 | 2 | 0 | 0 | 0 | 0 | 6 | 2 | 6 | 2 |

### 3.2. Evaluation of BaP Concentrations

Annual average BaP concentrations at the Godów station exceeded the annual limit of 1 ng·m$^{-3}$ each year by several times between 2011 and 2020 (Figure 14). The highest mean and median concentrations were achieved in 2011. In contrast, the lowest median BaP concentration was in 2018 and the lowest mean BaP concentration was in 2017. The average annual BaP concentration in 2020 at Věřňovice station was 7 ng·m$^{-3}$ (Figure 15). For the daily concentrations at the Věřňovice station for 2020, it was possible to evaluate the dependence of the average daily BaP concentrations on the daily wind types (Figure 16). The highest average BaP concentrations were at the wind direction from the eastern sector from the station, then at variable wind. E is the direction representing the transport from Poland to the Czech Republic.

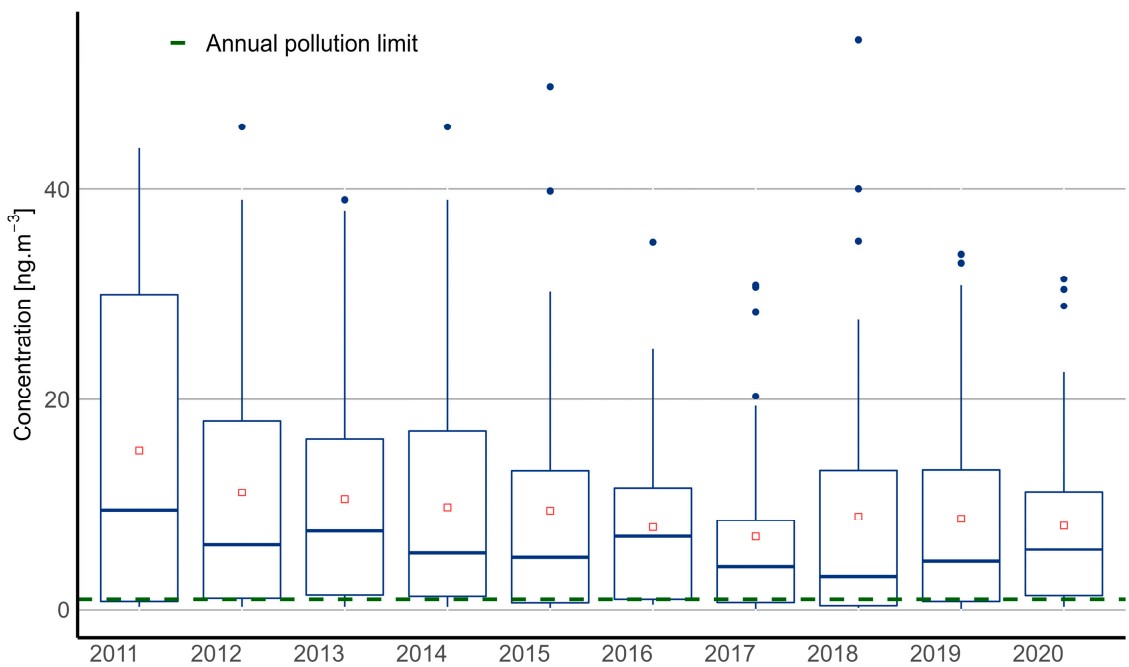

**Figure 14.** Boxplot of daily benzo[*a*]pyrene concentrations: Godów station, 2011–2020 (red squares indicate average concentrations; dashed line indicates the annual limit value for BaP).

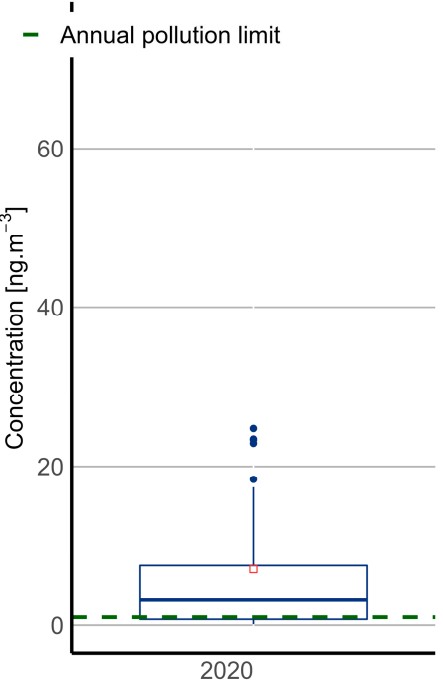

**Figure 15.** Boxplot of daily benzo[*a*]pyrene concentrations: Věřňovice station, 2020 (red squares indicate average concentrations; dashed line indicates the annual limit value for BaP).

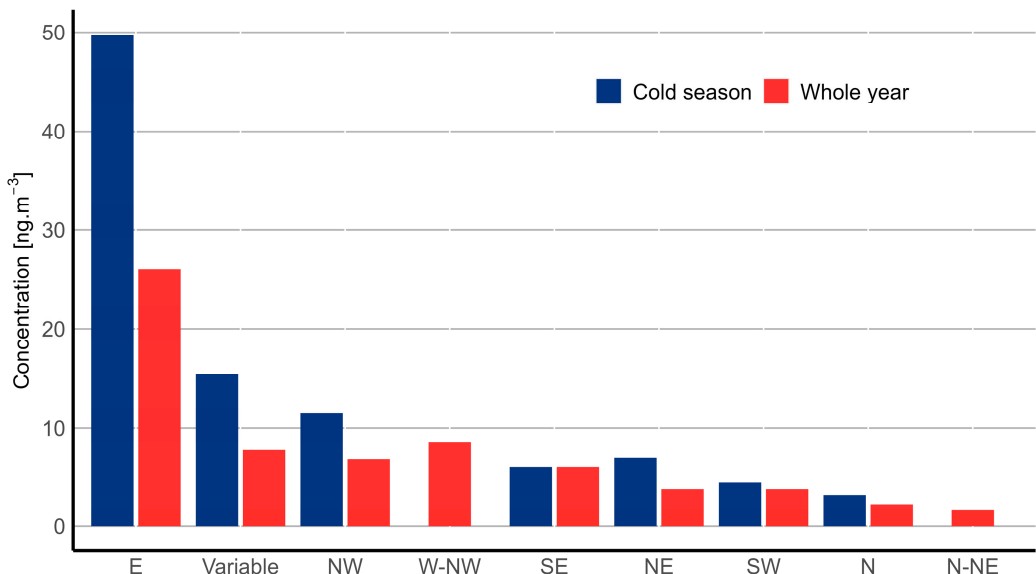

**Figure 16.** Average BaP concentrations divided into individual daily types of wind, Věřňovice station, 2020.

The highest BaP concentrations are at low temperatures below 1 °C. The average concentrations at temperature categories 1–5 °C and 5–10 °C are very similar and are about half the average BaP concentrations at temperatures below 1 °C. The lowest BaP concentrations are obtained at temperatures above 10 °C, but even these are on average higher than the annual BaP limit of 1 ng·m$^{-3}$ (Figure 17). Clearly, the highest average BaP concentrations are at wind speeds of 1 to 2 m·s$^{-1}$, roughly half that level at higher wind speeds above 2 m·s$^{-1}$, and the lowest average BaP concentrations are achieved at low wind speeds below 1 m·s$^{-1}$ (Figure 18).

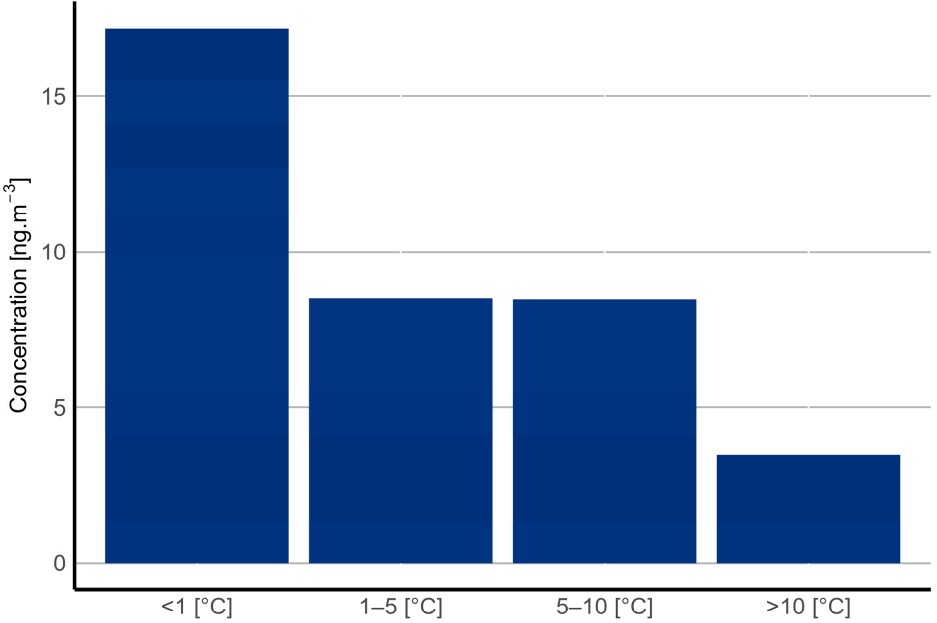

**Figure 17.** Average BaP concentrations by air temperature categories, Věřňovice station, 2020.

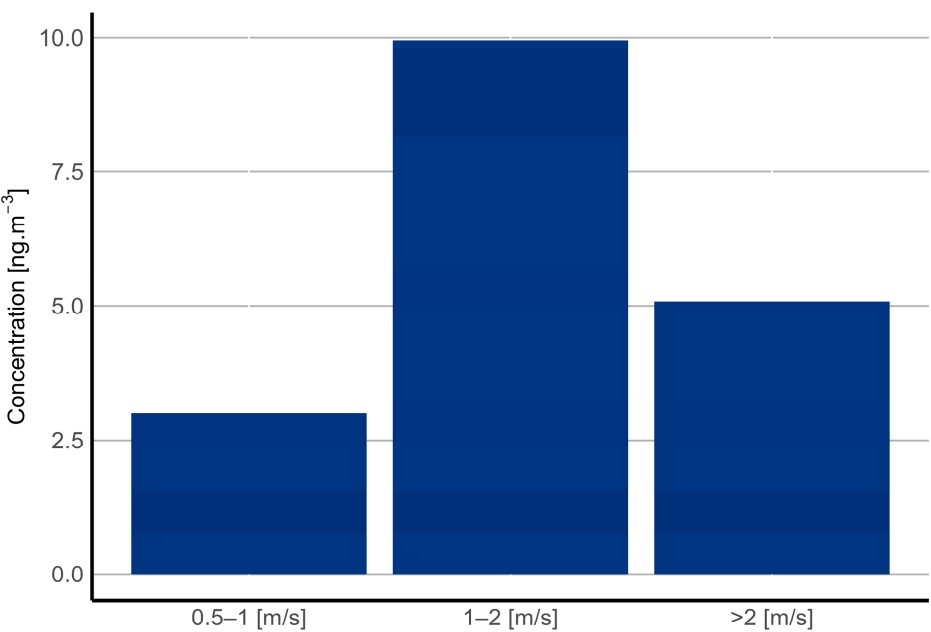

**Figure 18.** Average BaP concentrations by wind speed categories, Věřňovice station, 2020.

## 4. Discussion

This work confirms and refines the results from the previous work for the area of interest and complements the information from the Czech–Polish Air Silesia project from 2010–2013 [24,25], in which the transboundary transport was assessed only in co-dependence with $PM_{10}$ concentrations and for a short period (October 2009 to December 2012).

The possibility of assessing the transboundary air pollution transport of the Czech–Polish area has been further analysed in papers [36–41]. This paper deals with the evaluation of the relationships of $PM_{10}$ and BaP concentrations depending on meteorological conditions (mainly wind direction and wind speed) and relies only on measurements at ground stations at 2 m or 10 m above the ground (wind direction and wind speed) at stations Godów and Věřňovice. Transboundary transmission is also dealt with in a paper [41] prepared in the framework of the Air Border project [26]. The assessment is based on measurements at a height of 90 m on the former mining tower František in Horní Suchá in the Czech Republic. The František tower is located at a distance of about 12 km from the border with Poland in the NE direction. The measurements at this location of Horní Suchá at 90 m height are not representative for the Czech–Polish border area (Godów–Věřňovice area of interest) for the human breathing zone and pollutant measurements at ground stations. However, it provides valuable information on long-range transmission between the two countries and supports the assumption of higher transport of pollution from Poland to the Czech Republic.

Two stations were deliberately chosen for the evaluation, located at a short distance from each other and close to the border between the Czech Republic and Poland. The processing showed that the measurements at both stations are comparable in terms of magnitude and dependence on wind direction [24,25]. The dominant influence of the municipality of Věřňovice and the Dětmarovice power plant on $PM_{10}$ and BaP concentrations can be excluded as they lie in different directions from the stations, and measurements at these stations can be used to estimate the quantification of transboundary transport. To the east of the Věřňovice station are the Polish villages of Godów, Golkowice and Skrbeńsko with dense housing and local heating with solid fuels.

Quantification of the estimate of transboundary transport of BaP in the area of interest has not been undertaken, unlike for $PM_{10}$ (Tables 2–4). The authors would consider this information to be misleading given the insufficient data. Measurements of BaP concen-

trations in Věřňovice were carried out every 6th day in 2020. Thus, it was not possible to evaluate all meteorological situations during the year. Although there is a long series of BaP measurements at the Godów station, there is no indication to which day the concentration is related (https://powietrze.gios.gov.pl/pjp/archives, accessed on 10 January 2022). The daily values are broken down into days of the whole week, which makes it impossible to assess air pollution in detail in relation to meteorological conditions and transboundary transport between the two countries.

For the assessment of daily concentrations of pollutants, the CHMI uses an internal methodology for determining the daily type of wind direction. This method has also been used in this paper and in other treatments [24,25], but especially in unpublished air quality assessments where only 24-h average pollutant concentrations are available. Although the authors have studied enough papers on possible averaging of wind direction, and even familiarized themselves with some calculators for its calculation [42], they based their assessments on the determination of the prevailing daily wind direction by the methodology of the CHMI.

The problem is that the wind direction is usually given as an angle in degrees, 0–360 (or up to 359), where 0 (or 360) represents wind from a northerly direction. If the wind direction is from the north and passes through a discontinuity at the start/end of the circular scale, and then the arithmetic mean is calculated, the result will be an average wind direction somewhere in the southern quadrant, which is not correct. In order to properly resolve this scale discontinuity, trigonometric functions must be used to work with the angles. Wind should be viewed as a vector quantity (it has magnitude and direction) [43–47].

This quantification of transboundary transport is based on direct measurements in the border area of interest. While the authors are aware that other tools exist to assess the transboundary transport of pollutants, they consider the approach to assessing transport in the paper to be appropriate despite any potential biases and uncertainties with respect to data availability. Tracking high short-term concentrations using the HYSPLIT (Hybrid Single-Particle Lagrangian Integrated Trajectory) [48] could be a suitable complement, but this was not the focus of this paper. Another possible approach is model evaluations, e.g., Positive Matrix Factorization Model [40]. However, these model evaluations are dependent on a more detailed dataset (elemental analysis), which has not yet been developed in the border region of interest. Another suitable modelling tool is, e.g., the photochemical transport model CAMx [49,50], which has been used within the Czech Republic in the Air Quality Improvement Programme [35,36].

## 5. Conclusions

The paper presents an assessment of the transboundary transport of $PM_{10}$ and benzo[a]pyrene in the Czech–Polish border region. Quantification estimation is described for the section of the state border between Věřňovice and Godów, which represents the main area of air pollution transport between the Czech Republic and Poland. The work builds on previous work carried out within the Air Silesia project in 2010–2013. The assessment confirms the persistent air quality problems in the border area, where significant exceedances of BaP and $PM_{10}$ limits occur. The assessment of air quality and transboundary air pollution transport in this area helps to address measures to reduce emissions from pollution sources (on the Czech side, for example, the implementation of "boiler subsidies" to reduce emissions from individual heating with solid fuels replacement of old heating boilers).

In the area of interest, the prevailing wind direction in the cold months is from the Czech Republic to Poland (46%), in the warm months from Poland to the Czech Republic (41%), and in the average of the entire period 2011–2020 the share of wind direction between the countries is balanced. Slightly less but also significant is the variable wind. In the average years of the whole period 2011–2020, the transport of average $PM_{10}$ concentrations at the stations Godów and Věřňovice is higher from Poland to the Czech Republic by about 7%. In the cold months of the period 2011–2020, although the wind direction from the

Czech Republic to Poland is predominant, the transport of average $PM_{10}$ concentrations is higher from Poland to the Czech Republic by 14%. The average concentration of the cold seasons representing the transport from Poland to the Czech Republic is 78 $\mu g \cdot m^{-3}$, and in the opposite direction 44 $\mu g \cdot m^{-3}$. There is a transmission of $PM_{10}$ concentrations between large industrial and densely populated areas of the Czech Republic and Poland, where there is also a high proportion of domestic heating with solid fuels. However, the negative impact of air pollution from high daily $PM_{10}$ concentrations in the cold parts of the year is predominant from Poland to the Czech Republic. This claim is supported by the numbers of average daily $PM_{10}$ concentrations above 50 and 100 $\mu g \cdot m^{-3}$, which are higher from Poland to the Czech Republic by about 14% (for values above 50 $\mu g \cdot m^{-3}$) and 26% (for values above 100 $\mu g \cdot m^{-3}$), respectively. A high contribution to air pollution in the area is made by the variable wind that is typical of low wind speeds. The average $PM_{10}$ concentrations in the area under variable flow in the cold months of 2011–2020 are 80 $\mu g \cdot m^{-3}$. It can therefore be assumed that the high density of houses with predominantly individual heating with solid fuels on both sides of the border contributes significantly to air pollution. High average daily $PM_{10}$ concentrations at wind speeds above 1 $m \cdot s^{-1}$ for directions representing transport from Poland to the Czech Republic indicate long-range transport from more distant industrial areas of Poland (e.g., Katowice agglomeration), but also from dense housing with individual solid fuel heating. The direction from the SE (not included in the transboundary transport due to ambiguity) at wind speeds above 2 $m \cdot s^{-1}$ may indicate the influence of more distant industrial sources from the Czech Republic, specifically from the Třinec area.

The evaluation of short-term $PM_{10}$ concentrations at the Věřňovice station in the long term shows a higher contribution of air pollution from NE directions (from Poland) for the average $PM_{10}$ concentrations and for the maximum contributions of these 1-h concentrations.

The highest average BaP concentrations at the Věřňovice station in 2020 come from the eastern sector, where the Polish territory with low emission sources is located. The highest 24-h BaP concentrations are obtained at wind speeds of 1–2 $m \cdot s^{-1}$ and above 2 $m \cdot s^{-1}$, which is typical for transmission from more distant locations.

**Author Contributions:** Conceptualization, V.V.; methodology, V.V. and D.H.; formal analysis, V.V. and D.H.; investigation, V.V.; resources, V.V.; data curation, D.H. and V.V.; writing—original draft preparation, V.V., B.K. and R.S.; writing—review and editing, R.S. and V.V.; visualization, D.H. and V.V.; supervision, V.V. and B.K.; project administration, V.V. All authors have read and agreed to the published version of the manuscript.

**Funding:** The measurement of benzo[*a*]pyrene in Věřňovice in 2020 was subsidised from the budget of the Moravian-Silesian Region.

**Institutional Review Board Statement:** Not applicable.

**Informed Consent Statement:** Not applicable.

**Data Availability Statement:** The data from measurements in the Czech Republic were provided for the analysis by the Czech Hydrometeorological Institute (www.chmi.cz, Czech Republic, accessed on 10 February 2022). The measurement data in Poland were downloaded from the website of the Chief Inspectorate of Environmental Protection (https://powietrze.gios.gov.pl/pjp/archives, Poland, accessed on 10 February 2022).

**Acknowledgments:** Thanks to the Czech Hydrometeorological Institute (Czech Republic) for providing the data and to the Moravian-Silesian Region (Czech Republic, Ostrava) for special measurements in Věřňovice in 2020.

**Conflicts of Interest:** The authors declare no conflict of interest.

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
