# Peer review of "Transboundary Air Pollution Transport of PM10 and Benzo[a]pyrene in the Czech–Polish Border Region"

_atmosphere, doi:10.3390/atmos13020341_

Round 1

Reviewer 1 Report

GENERAL IMPRESSION

The work has an average level of relevance and novelty, it can hardly be useful for solving regional problems. 

The significance of the design carried out in this paper is not well explained relative to other important works published in this field. The authors should review, comment, and compare more works that are developed.

Section 3.1  (Evaluation of BaP concentrations) has limited reliability and high uncertainty and in my opinion in this case should not be part of a scientific paper (for example the period for Godowo is 2011-2020 and for Vernovice only 2020, which cannot be compared)

The conclusion should be sharpened, the section seems to be too long, some sentences can be avoided

DETAILED INFORMATION

  1. Figure 1 has low quality and is unreadable. Should be fixed or canceled as it is not necessary. Moreover, the description below in Figure 1 (line 47-48) indicates these two photos as left and right, which are actually above the other and not next to each other.
  2. Figure 2 – low quality and unreadable
  3. [line 167-168; “8 basic directions (N, NE, NW, W, W, SW)”] there are only 6 in the bracket.
  4. [line 198; “is more detailed in view of the facts described in Chapter 2.”] what is chapter 2, as there is no such section in the paper?
  5. [line 201; “In the cold part of the year,”] could you be more specific about what exactly period is it?
  6. [line 327-329] The description of figure 14 indicates the figure (left) and the figure (right) instead of the figure (up) and the figure (down)

Author Response

Dear reviewer,

thank you very much for your review, your comments and advice on corrections/additions. 
The opinion of all 4 reviewers is favourable. These are basically minor suggestions for corrections. Nevertheless, in some cases the reviews differ. We will try to accommodate all comments.
Comments/responses to your comments are provided in this cover letter and in the form of a change scheme directly in the article.

Reviewer 1:

Thank you very much for your review, your comments and advice on corrections/additions.

The opinion of all 4 reviewers is favourable. These are basically minor suggestions for corrections. Nevertheless, in some cases the reviews differ. We will try to accommodate all comments.

Comments/responses to your comments are provided in this cover letter and in the form of a change scheme directly in the article.

……………………………………

The work has an average level of relevance and novelty, it can hardly be useful for solving regional problems.

The elaboration (References 23-25) within the framework of the international AirSilesia project, to which this article is closely related, was the trigger for many negotiations between the Czech and Polish sides. On the basis of these findings, both sides have focused on a more detailed assessment of air quality in the area of concern and an assessment of transboundary transport. Taking into account that this is an area with a high population density and therefore a high risk to human health. The elaboration has become a mainstay for the regional public authorities (regional authorities and cities). For example, replacements of old boilers with new individual heating systems have been introduced. We believe that any treatment that highlights an air pollution problem in the area is important and serves as a basis for further (more in-depth/detailed) treatment (e.g. modelling). The evidence also serves for operational assessment of air quality in the area, e.g. in the declaration of smog situations and regulations and the evaluation of episodes with high concentrations of air pollutants.

The significance of the design carried out in this paper is not well explained relative to other important works published in this field. The authors should review, comment, and compare more works that are developed.

We added a paragraph and references to the Discussion section.

Section 3.1  (Evaluation of BaP concentrations) has limited reliability and high uncertainty and in my opinion in this case should not be part of a scientific paper (for example the period for Godowo is 2011-2020 and for Vernovice only 2020, which cannot be compared)

You're right. The assessment of BaP concentrations has limited predictive value and high uncertainty. Nevertheless, we would like to keep this part in the treatment. We have tried to describe all uncertainties and uncertainties (see Experiments, Discussion and Conclusion sections). Unfortunately, the biggest problem is the lack of BaP data assigned to a specific date. Although there are 10 years of measurements of BaP concentrations at Godow station, it is not possible to evaluate them in detail. At the Vernovice station, the data can be more detailed (despite the uncertainties mentioned, e.g., that it is not measured every day), but we only have 1 year of BaP measurements. This article could point out the lack of identifiability of daily BaP values on the Polish side (without identifying a specific day with measurements), or the lack of BaP measurement period on the Czech side in this border region.

We have split the original Figure 14 into 2 figures so as not to give the impression that we want to compare them with each other. The figures are mainly intended to highlight the high daily BaP concentrations and the above limit annual concentrations in the region of interest.

We have rewritten Section 3.1 to Section 3.2.

The conclusion should be sharpened, the section seems to be too long, some sentences can be avoided

Fixed. The conclusion has been significantly shortened.

DETAILED INFORMATION

  1. Figure 1 has low quality and is unreadable. Should be fixed or canceled as it is not necessary. Moreover, the description below in Figure 1 (line 47-48) indicates these two photos as left and right, which are actually above the other and not next to each other.

Fixed. Figures 1 were canceled.

  1. Figure 2 – low quality and unreadable

Fixed. Figure 2 was divided into 2 parts: figure 1 and figure 2.

  1. [line 167-168; “8 basic directions (N, NE, NW, W, W, SW)”] there are only 6 in the bracket.

Fixed

  1. [line 198; “is more detailed in view of the facts described in Chapter 2.”] what is chapter 2, as there is no such section in the paper?

Fixed. Section Experiments.

  1. [line 201; “In the cold part of the year,”] could you be more specific about what exactly period is it?

Added to the text (months October to March 2011 to 2020).

  1. [line 327-329] The description of figure 14 indicates the figure (left) and the figure (right) instead of the figure (up) and the figure (down)

Fixed. Figure 14 was divided into 2 parts: figure 14 and figure 15.

Most of the figures have been redrawn in higher resolution, with larger captions in the images for better readability.

………………………………

Reviewer 2 Report

This manuscript assessed the influence of PM10 and Benzo[a]pyrene in the Czech-Polish Border Region from Poland. The authors completely used the weather parameters to explain the potential influence of air pollution to Czech from Poland. My recommendation is to accept it after minor revision. My questions or suggestions are shown below.

  1. I suggested that the title can be modified: Transboundary Air Pollution Transport of PM10 and Benzo[a]pyrene in the Czech-Polish and Poland Border Region.
  2. Too many keywords. I suggested that the keywords should not be over 5.
  3. The Figures 1 and 2 are too small. The authors should modify the size for these two figures.
  4. The authors should enlarge the words for Figures 3, 4, 6, 7, 8, 12, and 14.
  5. The authors used air pollutant concentrations and weather data to investigate the influence of air pollution to Czech from Poland. How to exclude the influence of local pollution source in Czech? when the data suggested that air pollution in Poland is an important contributor to Czech only based on weather data.
  6. In line 123, please provide the sampling and analysis methods and QA/QC for BaP.
  7. The conclusion is too long. They should try to concentrate these descriptions.

Author Response

Dear reviewer,

thank you for your positive review and feedback.

We have tried to deal with all comments.

Comments/responses to your comments are provided in this cover letter and in the form of a change chart directly in the paper.

Reviewer 2:

Thank you for your positive review and feedback.

We have tried to deal with all comments.

Comments/responses to your comments are provided in this cover letter and in the form of a change chart directly in the paper.

……………………..

This manuscript assessed the influence of PM10 and Benzo[a]pyrene in the Czech-Polish Border Region from Poland. The authors completely used the weather parameters to explain the potential influence of air pollution to Czech from Poland. My recommendation is to accept it after minor revision. My questions or suggestions are shown below.

I suggested that the title can be modified: Transboundary Air Pollution Transport of PM10 and Benzo[a]pyrene in the Czech-Polish and Poland Border Region.

We would like to keep the original name which is a continuation of the Air Silesia project. We think that the name „The Czech-Polish border Region“ also includes the area „Poland Border Region“

Too many keywords. I suggested that the keywords should not be over 5.

We have reduced the number of keywords to 5 (Czech-Polish border; transboundary air pollution transport; PM10; benzo[a]pyrene; daily type of wind)

The Figures 1 and 2 are too small. The authors should modify the size for these two figures.

We deleted Figure 1 acording recommendation of another reviewer. We have split Figure 2 into 2 figures (Figure 1 and Figure 2). We have edited them and increased the resolution.

The authors should enlarge the words for Figures 3, 4, 6, 7, 8, 12, and 14.

The figures have been redrawn in higher resolution, with larger captions in the images for better readability.

The authors used air pollutant concentrations and weather data to investigate the influence of air pollution to Czech from Poland. How to exclude the influence of local pollution source in Czech? when the data suggested that air pollution in Poland is an important contributor to Czech only based on weather data.

Based on your comment, we have modified the wording in the conclusion to make it clear to readers that individual heating sources contribute to air pollution on both sides of the Czech-Polish border.

In line 123, please provide the sampling and analysis methods and QA/QC for BaP.

Information about the BaP measurement method has been added to the 2nd paragraph of the chapter ExperimentsThe conclusion is too long.

They should try to concentrate these descriptions.

The conclusion has been significantly shortened.

……………………………………………

Reviewer 3 Report

This paper deals with the study of PM10 and benzo[a]pyrene transboundary transport between Poland and Czech Republic in their border region, considering meteorological conditions. This research is important since there are not many studies of particles transboundary transport which allows decision makers to understand the pollution sources in different seasons. The analysis is well described, and its results are valuable, nevertheless some issues should be addressed before publication. My main concern is related with the BAP data. Some of those issues are listed below.

  1. Figure 1. In the first place, it is not understood why different years are presented for PM10 and BAP. On the other hand, this figure presents the Europe air quality, that is not a product of this research, then, the source of this image may be included, in addition, the study region must be presented to visualize the relationship of this map with the present study.
  2. Experiments. It is not explained how the PAH data from Godow were acquired. The website mentioned in Line 130 only goes up to 2019 and contains meteorological and PM2.5 data from the automatic station, but not BAP data, since these data requires detailed analysis. The origin of the BAP data is unclear. The way in which the analyzes were carried out, or in any case the source where their validity can be verified, must be included. In the case of Věřňovice station the situation is still darker.
  3. Results On page 126 is mentioned that “Although BAP was measured at Godow for the entire assessment period 2011–2020, it could not be fully included in the treatment”. Then, the results presented in Section 3.1 are not admissible at least a detailed explanation of those results as well as their correspondence sources. If the days on which the BAPs were determined are not known, then how was Figure 14 (up) constructed?

On page 131, authors stated that “Annual BAP concentrations can be calculated, but daily concentrations cannot be evaluated in relation to other variables, e.g. wind direction and wind speed” and in page 173 stated that “Measurements every 6th day do not objectively include situations and nor is it possible to include the occurrence of all types of meteorological conditions”.  Then it is not understood how Figure 14 (down) and Figures 15, 16 and 17 were constructed. This should be explained in detail.

  1. Conclusions. This section looks like a continuation of the discussion section. Some of the paragraphs can be included in the Discussion section, and conclusions could be more specific.

Author Response

Dear reviewer,

thank you for your reviews, comments and advice.

The opinion of all 4 reviewers is favourable. These are basically minor suggestions for corrections. However, in some cases the reviews differ. We will do our best to accommodate all comments to the satisfaction of all reviewers.

Comments/responses to your comments are provided in this cover letter and in the form of a change chart directly in the paper.

Reviewer 3:

Thank you for your reviews, comments and advice.

The opinion of all 4 reviewers is favourable. These are basically minor suggestions for corrections. However, in some cases the reviews differ. We will do our best to accommodate all comments to the satisfaction of all reviewers.

Comments/responses to your comments are provided in this cover letter and in the form of a change chart directly in the paper.

……………………………………..

This paper deals with the study of PM10 and benzo[a]pyrene transboundary transport between Poland and Czech Republic in their border region, considering meteorological conditions. This research is important since there are not many studies of particles transboundary transport which allows decision makers to understand the pollution sources in different seasons. The analysis is well described, and its results are valuable, nevertheless some issues should be addressed before publication. My main concern is related with the BAP data. Some of those issues are listed below.

  1. Figure 1. In the first place, it is not understood why different years are presented for PM10 and BAP. On the other hand, this figure presents the Europe air quality, that is not a product of this research, then, the source of this image may be included, in addition, the study region must be presented to visualize the relationship of this map with the present study.

Fixed. Figures 1 were canceled.

  1. Experiments. It is not explained how the PAH data from Godow were acquired. The website mentioned in Line 130 only goes up to 2019 and contains meteorological and PM2.5 data from the automatic station, but not BAP data, since these data requires detailed analysis. The origin of the BAP data is unclear. The way in which the analyzes were carried out, or in any case the source where their validity can be verified, must be included. In the case of Věřňovice station the situation is still darker.

We have added information on how the data was obtained (to the section Experiments).

The PM10 and BaP data are fully available on this site, for the entire processing period 2011-2020. Please check that you are using the correct link (https://powietrze.gios.gov.pl/pjp/archives) available. We and other reviewers can see the data page listed. It is possible that during your review the web page was just somehow edited/unavailable in its entirety (?).

  1. Results On page 126 is mentioned that “Although BAP was measured at Godow for the entire assessment period 2011–2020, it could not be fully included in the treatment”. Then, the results presented in Section 3.1 are not admissible at least a detailed explanation of those results as well as their correspondence sources. If the days on which the BAPs were determined are not known, then how was Figure 14 (up) constructed?

A description of the data used is given in the Experiments chapter. 

Daily average BaP concentrations at Godow station are not measured every day. The daily average value is not assigned to a specific day in the data (link), but is given for each day of the week.

It is not a problem to use daily concentrations to calculate boxplots for each year without identifying the specific day on which the concentration was measured.  Boxplots (1 value in a given week). A boxplot is a way of graphically visualizing numerical data using its quartiles. Boxplots show the differences between data sets without any assumptions of a normal distribution of the data, so they are non-parametric. The spacing between the elements of the middle part of the diagram indicates the degree of dispersion (variance) and skewness of the data.

Section 3.1 has been rewritten as Section 3.2.

  1. On page 131, authors stated that “Annual BAP concentrations can be calculated, but daily concentrations cannot be evaluated in relation to other variables, e.g. wind direction and wind speed” and in page 173 stated that “Measurements every 6th day do not objectively include situations and nor is it possible to include the occurrence of all types of meteorological conditions”.  Then it is not understood how Figure 14 (down) and Figures 15, 16 and 17 were constructed. This should be explained in detail.

The annual average can be calculated because the number of data meets the minimum number of daily data per calendar year to calculate the annual average. Boxplots can be created for the annual set of daily data. However, it is not possible for the daily BaP concentrations at the Godów station to be evaluated in relation to other variables since it is not known exactly which day this concentration was measured. this is only a problem for the Godów station.

The dependence of the daily BaP concentrations at the Věřňovice station can be evaluated as a function of meteorological variables, since here the data are directly attributed to the day of measurement. However, from the station Věřňovice we have only daily data from 2020, in other years of the period 2011-2020 BaP was not measured in Věřňovice.

The figures are therefore correct.

A description of the input data, including the uncertainties described, is given in the Experiments section.

  1. Conclusions. This section looks like a continuation of the discussion section. Some of the paragraphs can be included in the Discussion section, and conclusions could be more specific.

Fixed. The conclusion has been significantly shortened.

…………………………….

Reviewer 4 Report

Atmosphere (ISSN 2073-4433)

Manuscript ID:  atmosphere-1579993

Title:  Transboundary air pollution transport of PM10 and benzo[a]pyrene in the Czech-Polish border region

--------------

The topic of the manuscript is interesting and the authors had a good idea for a research project. The subject fall within the general scope of the journal. The title clearly reflect the contents. Authors should further emphasize on the novelty of their work.

The article occupies with the evaluation of transboundary transport of air pollutants (PM10 and benzo[a]pyrene) in the Czech-Polish border region in Europe. The authors used simple modeling of aerosol transport, but in this case it is properly selected. The manuscript contains several technical errors. After their removal, it is ready for publication.

Detailed comments:

Figure 1:   Cite the source of these figures. In my opinion, these are figures from the European Environment Agency. Make sure you have the right to use them in your publication. Does the EEA provide them under an open license (eg Creative Commons)? Use files with higher resolution for better readability.

Line 48 (Figure caption) & L76:    BaP - The first use of an abbreviation in the main text, abstract and figure captions should be explained.

L31 and others:  Correct the citation of publications according to the guidelines for the authors (applies to many places in the manuscript), eg. [1-5]

L123: Don't start your sentence with an abbreviation and a number. It should be:  Concentration of BaP ... or:   Benzo[a]pyrene concentration… Check the entire manuscript.

L126: It should be:  Godów

L146: Mark which figure is (a) and which (b). In Fig. 2a, I propose to show the greater part of Europe, so that readers outside Europe will find it easier to see where the measuring points are located. It does not have to be all of Europe. For example, it can show Central Europe and the Northern part. Highlight the borders of the Czech Republic and Poland for better readability of the map.

Figure 3 and others:  Please use a larger font in the figures for better readability (e.g. axis descriptions, legends).

L212 and others:  I recommend using the multiplication sign in the unit (µg∙m-3) instead of the regular dot. Applies to the entire text.

L469-573 (References): Order this section. Some lines are missing some source data, eg L528 - which means GIOS. Godów??  Check the citation rules in the authors guide.

Author Response

Dear reviewer,

thank you very much for your positive review and feedback.

We have solved all comments.

Comments/responses to your comments are provided in this cover letter and in the form of a change chart directly in the paper.

Reviewer 4:

Thank you very much for your positive review and feedback.

We have solved all comments.

Comments/responses to your comments are provided in this cover letter and in the form of a change chart directly in the paper.

…………………….

The topic of the manuscript is interesting and the authors had a good idea for a research project. The subject fall within the general scope of the journal. The title clearly reflect the contents. Authors should further emphasize on the novelty of their work.

The article occupies with the evaluation of transboundary transport of air pollutants (PM10 and benzo[a]pyrene) in the Czech-Polish border region in Europe. The authors used simple modeling of aerosol transport, but in this case it is properly selected. The manuscript contains several technical errors. After their removal, it is ready for publication.

 Detailed comments:

Figure 1:   Cite the source of these figures. In my opinion, these are figures from the European Environment Agency. Make sure you have the right to use them in your publication. Does the EEA provide them under an open license (eg Creative Commons)? Use files with higher resolution for better readability.

Fixed. Figures 1 were canceled

Line 48 (Figure caption) & L76:    BaP - The first use of an abbreviation in the main text, abstract and figure captions should be explained.

Fixed.

L31 and others:  Correct the citation of publications according to the guidelines for the authors (applies to many places in the manuscript), eg. [1-5]

Corrected throughout the article

L123: Don't start your sentence with an abbreviation and a number. It should be:  Concentration of BaP ... or:   Benzo[a]pyrene concentration… Check the entire manuscript.

Corrected throughout the article

L126: It should be:  Godów

Fixed

L146: Mark which figure is (a) and which (b). In Fig. 2a, I propose to show the greater part of Europe, so that readers outside Europe will find it easier to see where the measuring points are located. It does not have to be all of Europe. For example, it can show Central Europe and the Northern part. Highlight the borders of the Czech Republic and Poland for better readability of the map.

Both figures (originally 2 on the left and 2 on the right) have been edited and renumbered to Figure 1 and Figure 2.

Figure 3 and others:  Please use a larger font in the figures for better readability (e.g. axis descriptions, legends).

Fixed.

L212 and others:  I recommend using the multiplication sign in the unit (µg∙m-3) instead of the regular dot. Applies to the entire text.

Fixed.

L469-573 (References): Order this section. Some lines are missing some source data, eg L528 - which means GIOS. Godów??  Check the citation rules in the authors guide.

Fixed.

Round 2

Reviewer 3 Report

The authors have addressed all comments appropriately and the document has been improved. Then it is suitable for publication.

Reviewer 4 Report

Well done! now